# Morphological variation of the early human remains from Quintana Roo, Yucatán Peninsula, Mexico: Contributions to the discussions about the settlement of the Americas

Mark Hubbe [1,2]☯*, Alejandro Terrazas Mata [3]☯, Brianne Herrera [1], Martha E. Benavente Sanvicente [4], Arturo González González [5], Carmen Rojas Sandoval [6], Jerónimo Avilés Olguín [7], Eugenio Acevez Núñez [7], Noreen Von Cramon-Taubadel [8]

1 Department of Anthropology, Ohio State University, Columbus, OH, United States of America, 2 Instituto de Arqueología y Antropología, Universidad Católica del Norte, San Pedro de Atacama, Chile, 3 Instituto de Investigaciones Antropológicas, Universidad Nacional Autónoma de México, Ciudad de México, Mexico, 4 Laboratorio de Prehistoria y Evolución del Instituto de Investigaciones Antropológicas, Universidad Nacional Autónoma de México, Ciudad de México, Mexico, 5 Museo del Desierto AC, Saltillo, Mexico, 6 Instituto Nacional de Antropología e Historia, Ciudad de México, Mexico, 7 Instituto de la Prehistoria de América AC, Playa del Carmen, Mexico, 8 Department of Anthropology, State University of New York – Buffalo, Buffalo, NY, United States of America

☯ These authors contributed equally to this work.
* hubbe.1@osu.edu

## Abstract

The human settlement of the Americas has been a topic of intense debate for centuries, and there is still no consensus on the tempo and mode of early human dispersion across the continent. When trying to explain the biological diversity of early groups across North, Central and South America, studies have defended a wide range of dispersion models that tend to oversimplify the diversity observed across the continent. In this study, we aim to contribute to this debate by exploring the cranial morphological affinities of four late Pleistocene/early Holocene specimens recovered from the caves of Quintana Roo, Mexico. The four specimens are among the earliest human remains known in the continent and permit the contextualization of biological diversity present during the initial millennia of human presence in the Americas. The specimens were compared to worldwide reference series through geometric morphometric analyses of 3D anatomical landmarks. Morphological data were analyzed through exploratory visual multivariate analyses and multivariate classification based on Mahalanobis distances. The results show very different patterns of morphological association for each Quintana Roo specimen, suggesting that the early populations of the region already shared a high degree of morphological diversity. This contrasts with previous studies of South American remains and opens the possibility that the initial populations of North America already had a high level of morphological diversity, which was reduced as populations dispersed into the southern continent. As such, the study of these rare remains illustrates that we are probably still underestimating the biological diversity of early Americans.

**Data Availability Statement:** All relevant data are within the manuscript and its Supporting Information files.

**Funding:** The authors received no specific funding for this work.

**Competing interests:** The authors have declared that no competing interests exist.

# Introduction

The human settlement of and dispersion across the Americas has been one of the most debated topics in archaeology and biological anthropology, with hundreds of articles published about the topic in the last decade alone. The initial occupation of the Americas has spun so much interest because the continent was the last large landmass on the planet to be occupied by humans, with a significant gap between the occupation of the other continental landmasses (~35 kya for the last regions of the old world: north Europe and Asia) and the initial crossing of human groups into the Americas (~20–15 kyr, according to [1]; but see [2–5] for suggestions of earlier occupations dates in North and South America).

The large scholarly interest in the occupation of the Americas also derives in equal measure from the historic events that led to the European colonization of the New World and the impact that North American academia has had in defining the mainstream research agenda in archaeology around the planet during the last century. The colonial interest in the origins of the Native American populations dates back to the initial decades of European presence in the continent (e.g., [6]), when the unknown origins of Native Americans in a land that was not seen as the birthplace of humankind invoked discussion about the origins of local native groups. To a large extent, this initial discussion set the tone for the centuries that followed, as the origins of Native Americans became an important mystery to be solved, leading to the establishment of numerous research projects to address the topic. As the United States assumed a role of leadership in archaeological research in the 20th and 21st centuries, this initial interest was translated into a large research program dedicated to the study of the settlement of the continent (see [7] for an early example).

From the inception of the academic discussion on the settlement of the Americas, the most important questions pursued regarded the timing, routes and biological origins of the first Americans. These three questions (when, where, and who) can be considered the broadest and most basal questions in understanding the process of human dispersion into the continent, and yet there is still a lack of consensus and considerable debate surrounding their answers (see, for example [1, 8]). Most certainly, an important factor contributing to this lack of consensus is that we are simplifying complex human processes into models that are not capturing the complex dynamics of human groups as they entered and occupied the continent. While creating models of dispersion is essential for us to be able to define testable hypotheses about the occupation of the continent, this practice also resulted in the establishment of oversimplified and, consequently, unrealistic models for the settlement of the Americas.

Take for example the discussion about the biological origins of Native American groups, since this is the focus of the present article. Genetic approaches to the study of human variation have shown conflicting results over the past several decades, with vastly different models previously defended to explain the biological diversity of Native American populations over the past 15 thousand years. The study of the biological diversity of the early occupants of the Americas has been approached indirectly, through the analysis of craniofacial [9–16], linguistic [17–19], and archaeological evidence [20, 21], as well as directly, with the study of DNA among modern Native American groups [22–25] and ancient remains [23, 26–28]. Over the past several decades, studies defended a wide array of scenarios, including a single migration into the continent [24, 25], two discrete early migrations into the continent [11, 12, 29], three dispersion events over the Holocene [17, 30], continuous gene-flow with Asia over the Holocene [9, 15], and different combinations of these [16, 27, 31]. Moreover, studies have defended different models of human dispersion after the initial process of settlement (e.g., [14, 23, 31]). This myriad of different scenarios speaks strongly of limitations to our ability to reconstruct reliable models for the settlement of and human dispersion across the Americas. These

limitations result, in a large degree, from shortcomings inherent in our data, which result in hypotheses and models about the settlement of the Americas systematically underestimating the amount of biological diversity observed in the continent during the Holocene.

For genetic studies, these limitations usually refer to limited samples available for analysis and result in studies based on different datasets that defend significantly different scenarios (for a recent example, see [23] and [31]). For craniofacial studies, these limitations refer to the complex model of inheritance and development of the morphological phenotype, which is the result of stochastic inheritance [32, 33] combined with responses to specific environmental and developmental pressures (e.g., [34]). Nonetheless, the study of craniofacial variation has for a long time been suggesting that there is considerable biological diversity in the Americas over time [10, 12–14, 16, 35], something that has only recently started to be identified in the genetic studies of current and past groups on the continent (e.g., [23, 31]). Taken together, these studies indicate that we still do not have an accurate picture of the biological diversity in the Americas over time, and until we have a better understanding of this diversity, it will be impossible to create reliable models for the settlement of the Americas.

Here, we contribute to the study of biological diversity in the Americas through the analysis of a series of early fragmented skulls from the Quintana Roo region, Mexico. The Quintana Roo material is uniquely important for this discussion for several reasons: First, it represents some of the earliest human remains in the Americas (e.g., [36, 37]), dating to the final millennia of the Pleistocene and beginning of the Holocene. Second, their preservation is among the best in North America, representing the most abundant material available to study biological diversity in the northern continent. While early Holocene remains are more frequent in South America [38, 39], early human remains in North America are notoriously rare [40]. And finally, the Mexican territory represents a geographical funnel, connecting North to Central and South Americas, and as such probably played an important role in the dispersion process between the northern and southern continents. Indeed, several studies have shown very high levels of craniofacial diversity among late Holocene Mexican groups [35, 41], suggesting that the region retained high levels of biological diversity until the end of the Holocene. Therefore, we aim to contribute to the discussion about the settlement of the Americas by testing whether early Mexican populations fit easily in our current understanding of the biological diversity of early American populations.

## The early human remains from Quintana Roo

The Quintana Roo subterraneous karst system is among the most extensive active cave systems worldwide (Fig 1), with a presumed extension of 700 km or more [42]. The cave system was carved mostly during the Pleistocene [43] by a series of sea level oscillations and changes in the overall hydrology, which intermittently exposed large parts of the cave system. When sea level rose at the end of the Pleistocene and the Early Holocene, between 13 and 7.6 kyr BP, this enormous karst labyrinth was flooded for the last time, preserving both archaeological and anthropological information [43].

At least eight sites with human skeletal remains dating to the Pleistocene-Holocene transition (13–8 kyr BP) have been identified in the Tulum area of Quintana Roo (Fig 1). These sites range from a few hundred meters to a maximum of 10 km from the current coastline [36, 44–46]. The human skeletal remains in these sites were found in depths ranging from a few meters to over 40m of the submerged cave systems. These individuals were almost certainly deposited in their location before the caves were submerged, and as such are considered to have been in situ throughout the whole period in which the caves have been flooded. This information is important, as it ascertains the antiquity of the remains included in this study, and is supported

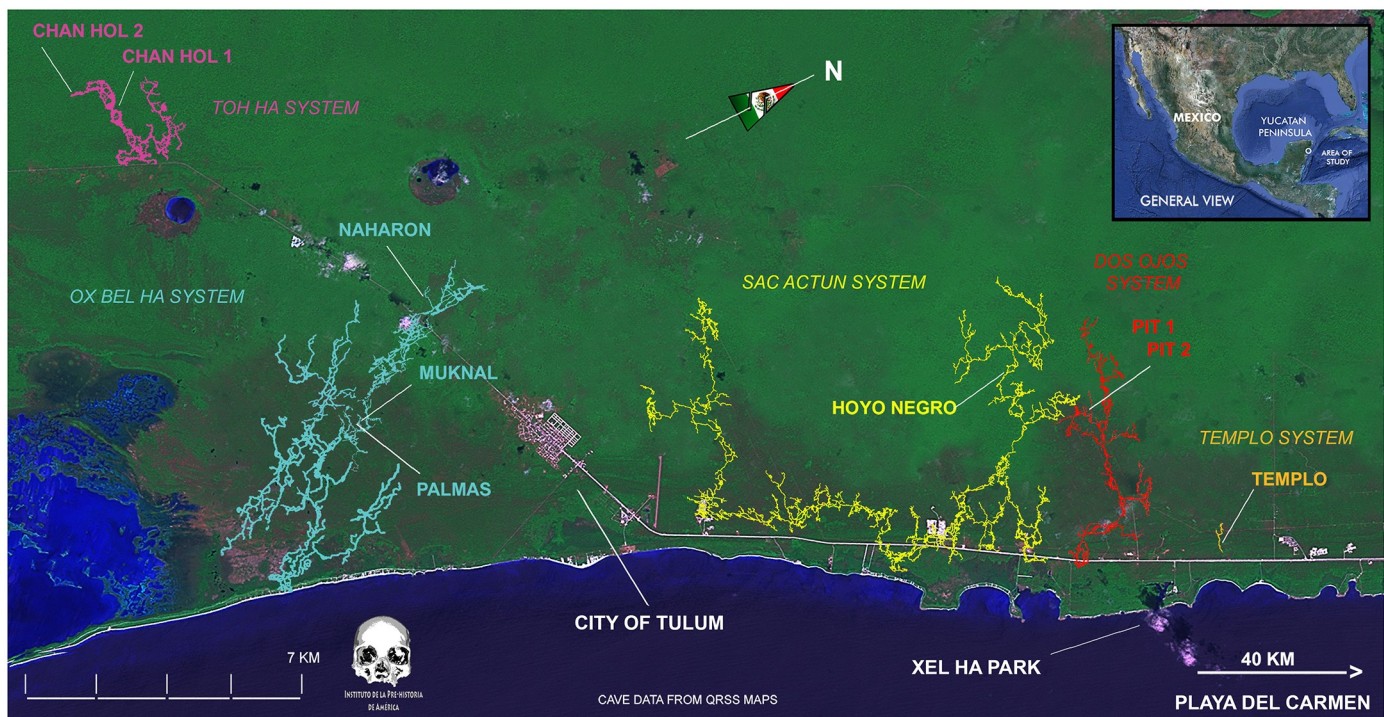

**Fig 1. Coast of the Mexican State of Quintana Roo with location of cenotes and caves containing sites with human skeletons and associated Pleistocene fauna.**
The area is presently restricted to a 20 km North-South directed stretch close to Tulum and extends towards Playa del Carmen. All sites are between a few hundred meters to a maximum of 10 km from the recent coastline. Map created by JAO, using satellite image from USGS. Maps of the caves are available in [13].

by several complementary pieces of evidence: (a) the human skeletons discovered at Naharón, Las Palmas, Chan Hol I and II, El Templo, Muknal and El Pit (Fig 1) were discovered in deep parts of the caves, which were flooded during early stages of sea-level rise, and are located hundreds of meters away from the nearest modern sinkhole (cenote); (b) four human skeletons (El Templo, Las Palmas, Chan Hol I and II) were almost fully articulated (including carpals and tarsals) and almost complete (>80% of bones represented), without major bone displacement. This situation clearly indicates an *in situ* decay of the bodies, likely while the cave was still dry. And (c) the flexed positions of the Naharón, Las Palmas, Chan Hol I and Chan Hol II individuals suggest intentional deposition of the human remains, adding support to the hypothesis that the caves were dry at the time. Intentional placement of human bones is also indicated for the Muknal site discovered at 30 m water depth [47] (see also SI1). For the other individuals (El Pit, Chan Hol and El Templo), however, this situation is less clear and the final position of these skeletons appears to be the result of either accidental death in a cave (El Templo), or spreading as a result of intermittent floating in the water (El Pit; [44, 46]), which is similar to Black Hole site, documented recently by Chatters et al. [36].

The four crania from Quintana Roo included in this study (Fig 2; S1 Text) have been dated to the end of the Pleistocene/ beginning of the Holocene (Table 1). The absolute dating was accomplished by using different radiometric techniques (AMS and U/Th), both on bone and on charcoal. However, AMS dates on human bones must be taken with caution, as the amount of preserved collagen is very small in these cases [48], which can affect the accuracy of the date. For example, the Las Palmas individual was dated to 8,050+/-130 BP using AMS, and to 12,000–10,000 BP using U/Th techniques, illustrating the range of possible error. The Charcoal sample from Muknal was collected from inside the skull and is assumed to be contemporary

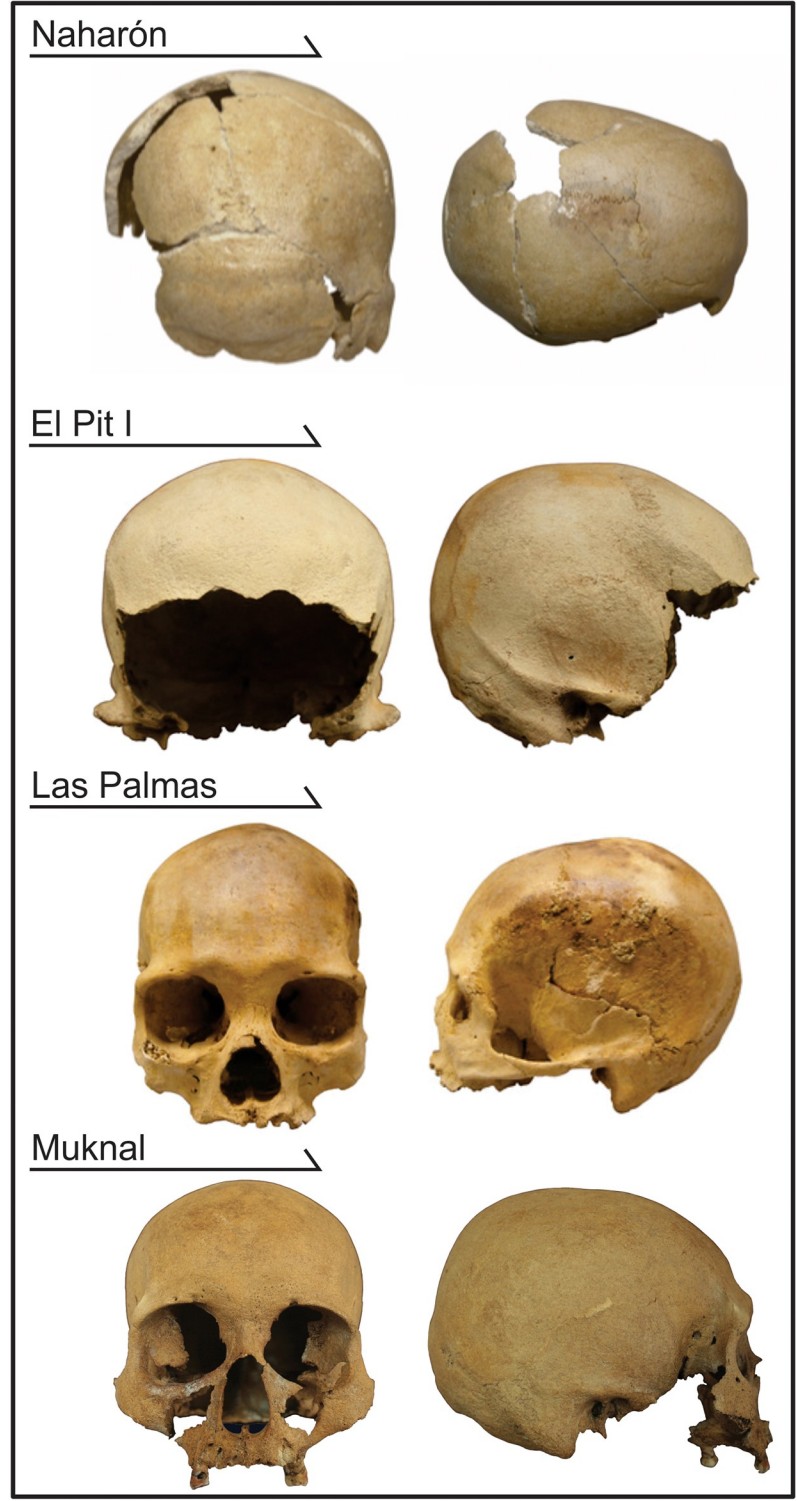

**Fig 2. The early Quintana Roo specimens analyzed in this study.**

**Table 1. Radiometric dates of human bone and charcoal associated with the skeletons from submerged caves in the Tulum area, Quintana Roo, Mexico.**

| Lab. Number | Individual | Sample | Laboratory | Radiocarbon Age BP ±1σ | Calibrated BP 2σ[1] |
|---|---|---|---|---|---|
| UCR4000A/CAMS-87301) | Naharón | [14]C AMS total amino acids from human bone | University of California Riverside | 11,670±65 | 13,277–13,499 |
| KIA435224 | El Pit 1 | [14]C on apatite from human bone | Kiel | 11,332±64 | 13,073–13295 |
| UGA6828 | Las Palmas | [14]C AMS collagen from human bone | University of Georgia | 8,050±130 | 8,587–9,306 |
| Las Palmas/Oxford | Las Palmas | U/Th on human bone | Oxford University | | 10,000–12,000 |
| UNAM1240 | Muknal | [14]C on charcoal | LUR-UNAM | 8,980±100 | 9,731–10290 |

[1] Calibration according to CALIV REV7.1.0 using intcal13.14c.

or slightly younger than the bones, due to its preservation (it shows no evidence of weathering or degradation). The dating of the charcoal suggests an age older than 11,011 ± 133 BP ([47]; see also [44, 46] for additional charcoal dating of alleged fireplaces inside the caves). Therefore, there is strong support for the individuals of Quintana Roo to represent populations that occupied the area between the end of the Pleistocene and the beginning of the Holocene.

A comprehensive description of the human material from the Yucatan submerged caves can be found in González et al. [44, 46] and Terrazas et al. [49]. Four individuals (Naharón, Las Palmas, El Pit I, Muknal) have crania sufficiently well preserved, and were included in the morphometric analyses presented here. For additional morphological information regarding this later individual, see Terrazas et al. [49] (see also [44–46]). The specific context of each of these individuals is also provided as Supplementary Information (S1 Text).

## Materials and methods

In this study, we explore the morphological affinities of four crania recovered from the caves in Quintana Roo (Fig 2). The individuals come from Naharón (Specimen number: PAC2002-1 (Na) B1a753), Las Palmas (specimen number: PAC2002-PALMAS-H-1), Muknal (PQR2011-PALMAS-H-2) and El Pit (Individual 1) (PQR2011- JACINTO PAT-1). All specimens are housed in the Laboratorio Arqueológico Xochimilco, Vivero Netzahualcóyotl, Calle Leandro Valle S/N, Col. Ciénega Grande, Delegación Xochimilco, Ciudad de México. All necessary permits were obtained from the Consejo de Arqueología of the Instituto Nacional de Antropología e Historia (No. C.A.401-36/0960).

The skulls were CT-scanned and Type I and II landmarks representing their craniofacial morphology were collected from each individual by one of us (BH). The complete protocol of landmarks used for this study includes 76 type I and type II landmarks. However, the different states of preservation of the specimens precluded the use of the complete landmark set for the analyses. The number of landmarks collected for each individual is listed in Table 2 and range from 11 to 37 landmarks. Their values are reported in S1 Table. Unfortunately, there is only a small number of landmarks that are common across all specimens, so in this study we compare each specimen to the reference dataset individually. This limits our ability to explore the affinities among the Quintana Roo specimens, but such a comparison is not statistically feasible due to the lack of common landmarks across specimens. For similar reasons, there was no attempt to estimate missing values in the dataset, as the specimens are too few and incomplete to be used to infer missing landmarks with reliability. Given the problems derived from dealing with individual specimens (see below), the inference of landmarks would also potentially add another source of error for the analysis of morphological affinities.

**Table 2. Landmarks available in each of the Quintana Roo specimens.**

| Landmark | ID[1] | Naharón | El Pit | Las Palmas | Muknal |
|---|---|---|---|---|---|
| Inion | 1 | X | X | X | X |
| Asterion R | 2 | X | X | X | X |
| Asterion L | 3 | | X | X | X |
| Lambda | 4 | | X | X | X |
| Basion | 5 | | | X | |
| Opisthion | 6 | | | X | |
| Hormion | 9 | | | X | |
| Stylomastoid Foramen R | 11 | X | | X | X |
| Porion R | 16 | X | X | X | X |
| Lat. Glenoid R | 19 | X | X | X | X |
| Frontomalare Posterior R | 22 | | | X | X |
| Stylomastoid Foramen L | 23 | | | X | X |
| Porion L | 28 | | X | X | X |
| Lat. Glenoid L | 31 | | X | X | X |
| Zyg-temp Suture Inf L | 32 | | | X | |
| Zyg-temp Sututre Sup L | 33 | | | X | |
| Frontomalare Posterior L | 34 | | | X | X |
| Bregma | 50 | X | X | X | X |
| Glabella | 51 | X | | X | X |
| Nasion | 52 | | | X | X |
| Orbitale Superior Right | 57 | X | | X | X |
| Dacryon R | 58 | | | X | X |
| Orbitale R | 59 | | | X | X |
| Zygoorbitale R | 60 | | | X | X |
| Frontomalare Orbitale R | 61 | | | X | X |
| Zygomaxillare R | 63 | | | X | |
| Alare R | 64 | | | X | X |
| Jugale R | 65 | | | | X |
| Stephanion R | 66 | X | X | X | |
| Orbitale Superior Left | 68 | X | | X | X |
| Dacryon L | 69 | | | X | X |
| Orbitale L | 70 | | | X | |
| Zygoorbitale L | 71 | | | X | |
| Frontomalare Orbitale L | 72 | | | X | X |
| Zygomaxillare L | 74 | | | X | |
| Alare L | 75 | | | X | X |
| Jugale L | 76 | | | X | X |
| Stephanion L | 77 | X | X | X | |
| Total | | 11 | 11 | 37 | 27 |

[1]–Landmark ID indicates the landmark number for the raw data in S1 Table.

The morphological affinities of the Quintana Roo specimens were assessed by comparing them to a reference sample of worldwide modern human populations [16, 50], comprised of 18 population samples (Table 3). The comparative dataset was collected by NvCT and has been used in previous studies exploring the morphological affinities of Early South American samples from Lagoa Santa, Brazil [16]. To date, it represents one of the largest comparative

**Table 3. Human population craniometric samples used as reference samples.**

| Population | Region | N | Lat, Long | Museum[1] |
|---|---|---|---|---|
| San | Africa | 31 | -21.0, 20.0 | NHM, MH, AMNH, NHMW, DC |
| Biaka | | 21 | 4.0, 17.0 | NHM, MH |
| Ibo | | 30 | 7.5, 5.0 | NHM |
| Zulu | | 30 | -28.0, 31.0 | NHM |
| Berber | | 30 | 32.0, 3.0 | MH |
| Italian | Europe | 30 | 46.0,10.0 | NHMW |
| Basque | | 30 | 43.0, 0.0 | MH |
| Russian | | 30 | 61.0, 40.0 | NHMW |
| Australian | Australo-Melanesia | 30 | -22.0, 126.0 | DC |
| Andaman | | 28 | 12.4, 92.8 | NHM |
| Mongolian | Asia | 30 | 45.0,111.0 | MH |
| Chinese | | 30 | 32.5,114.0 | NHMW |
| Japanese | | 30 | 38.0,138.0 | MH |
| Alaskan | Arctic North America | 30 | 69.0, -158.0 | AMNH |
| Greenland | | 30 | 70.5, -53.0 | SNMNH |
| Hawikuh | Americas | 30 | 33.5, -109.0 | SNMNH |
| Chubut | | 30 | -43.7, -68.7 | MLP |
| Lagoa Santa | Early America | 30 | -19.4, -44.0 | ZMD, RIO, BH, USP |

[1] NHM, Natural History Museum (London, UK); MH, Museé de l'Homme (Paris, France); AMNH, American Museum of Natural History (NY, USA); NHMW, Das Naturhistorische Museum, Wien (Vienna, Austria); DC, Duckworth Collection (Cambridge, UK); SNMNH, Smithsonian National Museum of Natural History (Washington, D.C., USA); Museo de la Plata (La Plata, Argentina); ZMD, Zoological Museum, University of Copenhagen (Denmark); RIO, National Museum, Federal University (Rio de Janeiro, Brazil); BH, Museu de História Natural, Federal University of Minas Gerais (Belo Horizonte, Brazil); USP, University of São Paulo (Brazil).

datasets for 3D craniofacial landmarks, and the only one that includes a reference sample from early Americans. All early Americans included in this series come from the region of Lagoa Santa, which represents the largest collection of early Holocene skulls in the Americas (details about the collection can be found in [11,16]).

The 3D landmarks for each of the Quintana Roo specimens were combined with the reference samples, and each of the final datasets was processed through Generalized Procrustes Analysis (GPA) to remove the effect of size, rotation and translation between specimens [51]. The data post-GPA was transformed into Principal Components, by decomposing the total covariance matrix into its eigenvalues and eigenvectors and rotating the original data according to the coordinates of the eigenvectors [52]. The transformation into Principal Components is important because it transforms the original 3D coordinates into scaled orthogonal variables, that are not correlated (i.e., they have variances = 1 and covariances = 0) and that concentrate most of the explanatory power of the data into fewer variables. That is, the first Principal Component explains the largest amount of variance present in the original data, and so forth. Moreover, the transformation into Principal Components is an essential step to compare the morphological affinities of individual specimens to the values of population centroids.

Comparing isolated specimens to reference series is not straightforward, since it is impossible to know a priori if the isolated specimens represent the average shape of its original population (i.e., if the specimen is close in shape to the morphological average of the population), or if it is an outlier in that population. This posits a serious problem in the analysis of individual specimens, as the relationship of affinities observed may not be representing the true population biological affinities of the specimens. However, transforming the original data into

Principal Components and working with the most informative ones largely solves this problem, because the individual deviations from the population centroid will tend to be relegated to the less informative Principal Components. The morphology of any individual in a population can be described as the morphology of the group centroid (the average morphology of the population) plus an individual error. This error component, because it is unique to each individual, will tend to be relegated to the less informative Principal Components since the individual error has a small contribution to the overall shared variance in the data.

Therefore, to integrate individual specimens into larger comparative datasets, focusing on the most informative PCs is an effective solution to minimize the impact that the individual error has on the assessment of a specimen's morphological affinities. Evidently, this is only true as long as the population from which the individual comes is represented within the morphological variance of the reference dataset. This is a reasonable assumption in our case, as the comparative dataset represents the overall morphological diversity of modern humans.

Following this rationale, for each of the analyses we chose the number of PCs that explained around 50% of the original variance in the dataset. Since each specimen has a different number of landmarks and they were compared to the reference dataset individually, the final number of PCs used for each specimen varied, as indicated in Table 3. The morphological affinities were explored through three complementary analyses. The analyses were based on Mahalanobis distances ($D^2$) between series calculated from the Principal Components selected for each specimen (Table 3). Mahalanobis distances were used due to the prevalence of this distance in studies of morphological affinities (e.g., [10, 12, 35, 53]). However, it must be noted that in this case, $D^2$ is the same as common Euclidean distances since the former corrects the contribution of each variable based on the variance/covariance matrix, which in this case is an identity matrix (all PCs have variance of 1 and covariances of 0). The first analysis consisted of the Multidimensional Scaling (MDS) of the $D^2$ matrix, which generates the graphic representation of distances without assuming hierarchical relationships between them [52]. The goodness of fit for the MDS was calculated through the Kruskall's measurement of Stress, which informs how much the distance matrix is being deformed to be represented in the number of dimensions (two in this case) of the MDS solution. Low stress levels indicate better fit between the MDS solution and the distance matrix, and stress levels equal or below to 10% are usually considered be fair representations of the distance matrix [53]. Secondly, we analyzed the $D^2$ in a cluster analysis using Ward's algorithm of aggrupation [54]. Ward's method has been used in the past for studies about the morphological affinities of modern humans [10, 12, 55] and shows high consistency in the morphological affinities observed among populations in a global context. Finally, the third analysis consisted of the classification of the specimens into the reference populations, using both Posterior Probabilities and Typicalities [52]. The difference between these two measurements is that the former calculates the probability of an individual belonging to any of the reference samples assuming that it must belong to at least one of them, while the latter allows for the possibility that the individual can be considered as not belonging to any of the reference samples. The combination of the two probability measurements allows for the analysis of which is the closest population to the specimen and how likely the individual is to belong to that population.

Males and females were analyzed together, as the sexual dimorphism after size is controlled for (through GPA analysis) has been shown to be a non-significant source of variance, when compared to the scale of differences between populations [10, 13, 16, 56]. Despite the fact that the sex of some of the Quintana Roo specimens was able to be estimated from the skeletal remains, restricting the analysis to only males or females would reduce the sample size in most of the reference populations, which would add larger sources of error than the one generated

by grouping sexes. All analyses were performed in R [57] with functions written by MH and complemented by the packages geomorph [58], vegan [59], and MASS [60].

## Results

### Naharón

The cranial remains from Naharón (Fig 2) are estimated to be from a young adult female (see SI 1 for details) and represent the oldest of the skeletal remains analyzed here (cal BP 13,277–13,499; Table 1). The cranial remains from Naharón are largely fragmented, representing only the calvaria (frontal, portions of the parietals, and most of the occipital) and only 11 landmarks that matched the reference dataset could be collected from this individual. The analysis of morphological affinities for Naharón were based on the first two principal components, which explain 59.2% of the variance present in the dataset.

Fig 3a shows the result of the Multidimensional Scaling, which illustrates the morphological affinities of the series represented in the $D^2$ distance matrix. Because the Mahalanobis distances are based on only two dimensions (PCs), the MDS shows an almost perfect fit to the distance matrix (Stress = 0.008%). The plot shows worldwide patterns of clustering that have been previously described in the literature [61, 62]: there are clusters for each of the main continental regions reflected in the dataset (Africa, East Asia, Europe). Different from previous studies [10, 56, 63], in this analysis the Early Americans series of Lagoa Santa appears in the center of the plot, closer to other series from South America (Chubut), East Asia (Mongolia), and Australia. Interestingly, Naharón appears closely associated to arctic North American series (Greenland and Alaska), which have been described previously as robust cold adapted populations and quite distinct from Native Americans [9, 15, 64]. Previous studies have also found association between Early Americans and the arctic series [16, 64]. The Ward's Cluster generated for Naharón (Fig 3b) corroborates the associations seen in the MDS, with Naharón and the arctic populations appearing as an outlier to the rest of the groups in the dataset. The classification of Naharón is shown in Table 4. As can be seen, the highest posterior probabilities are between Naharón and Greenland (p = 0.319) and Alaska (p = 0.252). The Typicality results show Naharón to be very close to the centroid of these arctic populations (p = 0.974 and p = 0.771, respectively). However, a cautionary note must be added to the interpretation of the morphological affinities observed here, as they are based on a very small number of anatomical landmarks.

### El Pit I

The cranial remains from El Pit I (Fig 2) are also highly fragmented, with only the calvaria preserved enough for analysis. The individual has been dated to a similar time period as Naharón (cal BP 12,073–13,295; Table 1). El Pit I is estimated as a probable male and possibly died in the early stages of adulthood (see a complete description in SI 1). Only 11 of the landmarks available in the comparative dataset could be collected from this specimen.

The morphological affinities between El Pit I and the reference series are based on the first two Principal Components, which explain 55.0% of the variance in the dataset. Its relationship to the series is represented graphically through the MDS analysis (Fig 4a) and Ward's Cluster (Fig 4b). As in the case for Naharon, the MDS show an almost perfect representation of the Mahalanois distances (Stress = 0.01%). The same regional patterns of morphological affinities observed before are clear in these analyses. However, El Pit I shows stronger morphological affinities with European populations, which is a pattern of association not previously seen between early Americans and reference series (although Kennewick Man was initially described as sharing strong morphological affinities with Ainu, Polynesian and European

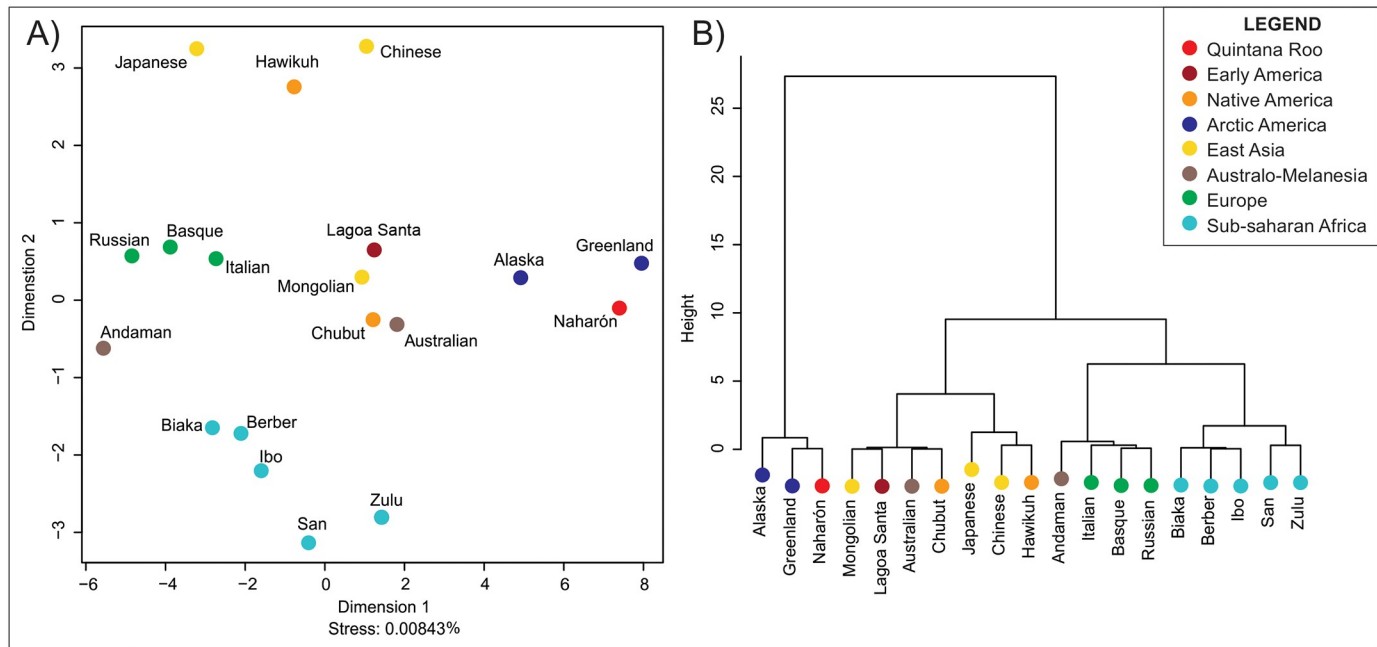

**Fig 3. Morphological affinities between Naharón and reference series according to Mahalanobis distances of the first two Principal components.** A) Multidimensional Scaling. B) Ward's Cluster.

**Table 4. Posterior probabilities and typicalities calculated for the classification of each of the Quintana Roo specimens into the reference series.**

| Reference series | Region | Naharón | | El Pit I | | Las Palmas | | Muknal | |
|---|---|---|---|---|---|---|---|---|---|
| | | Posterior Prob[1] | Typicality[2] | Posterior Prob[1] | Typicality[2] | Posterior Prob[1] | Typicality[2] | Posterior Prob[1] | Typicality[2] |
| San | Africa | 0.018 | 0.054 | 0.038 | 0.217 | 0.001 | *0.024* | <0.001 | *<0.001* |
| Biaka | | 0.004 | *0.012* | 0.089 | **0.514** | 0.017 | 0.166 | <0.001 | *<0.001* |
| Ibo | | 0.009 | *0.028* | 0.051 | 0.292 | 0.031 | 0.252 | <0.001 | *<0.001* |
| Zulu | | 0.055 | 0.168 | 0.009 | 0.050 | 0.006 | 0.080 | <0.001 | *<0.001* |
| Berber | | 0.007 | *0.021* | 0.047 | 0.269 | 0.019 | 0.184 | <0.001 | *<0.001* |
| Italian | Europe | 0.004 | *0.014* | 0.123 | **0.709** | 0.009 | 0.111 | 0.001 | *0.007* |
| Basque | | 0.002 | *0.005* | **0.158** | **0.910** | 0.003 | *0.041* | <0.001 | *0.003* |
| Russian | | 0.001 | *0.002* | **0.146** | **0.838** | 0.017 | 0.172 | 0.004 | *0.016* |
| Australian | Australo-Melanesia | **0.089** | 0.271 | 0.025 | 0.142 | 0.006 | 0.077 | <0.001 | *<0.001* |
| Andaman | | <0.001 | *0.001* | **0.125** | **0.719** | 0.102 | 0.522 | 0.001 | *0.003* |
| Mongolian | Asia | 0.056 | 0.172 | 0.025 | 0.144 | 0.003 | *0.045* | 0.097 | 0.190 |
| Chinese | | 0.038 | 0.115 | 0.014 | 0.080 | 0.066 | 0.408 | 0.084 | 0.171 |
| Japanese | | 0.002 | *0.006* | 0.102 | **0.584** | **0.111** | **0.546** | **0.106** | 0.201 |
| Alaska | Arctic North America | **0.252** | **0.771** | 0.001 | *0.007* | 0.053 | 0.356 | **0.225** | 0.329 |
| Greenland | | **0.319** | **0.974** | <0.001 | *<0.001* | 0.003 | *0.042* | 0.063 | 0.140 |
| Hawikuh | Americas | 0.014 | *0.043* | 0.020 | 0.117 | 0.013 | 0.142 | 0.053 | 0.124 |
| Chubut | | 0.067 | 0.204 | 0.018 | 0.101 | **0.196** | **0.717** | **0.317** | 0.406 |
| Lagoa Santa | Early America | 0.065 | 0.198 | 0.010 | 0.057 | **0.345** | **0.882** | 0.048 | 0.115 |

[1]–The three highest posterior probabilities for each group are highlighted in bold.

[2]–Typicalities above 0.5 are highlighted in bold; typicalities below 0.05 are highlighted in italics.

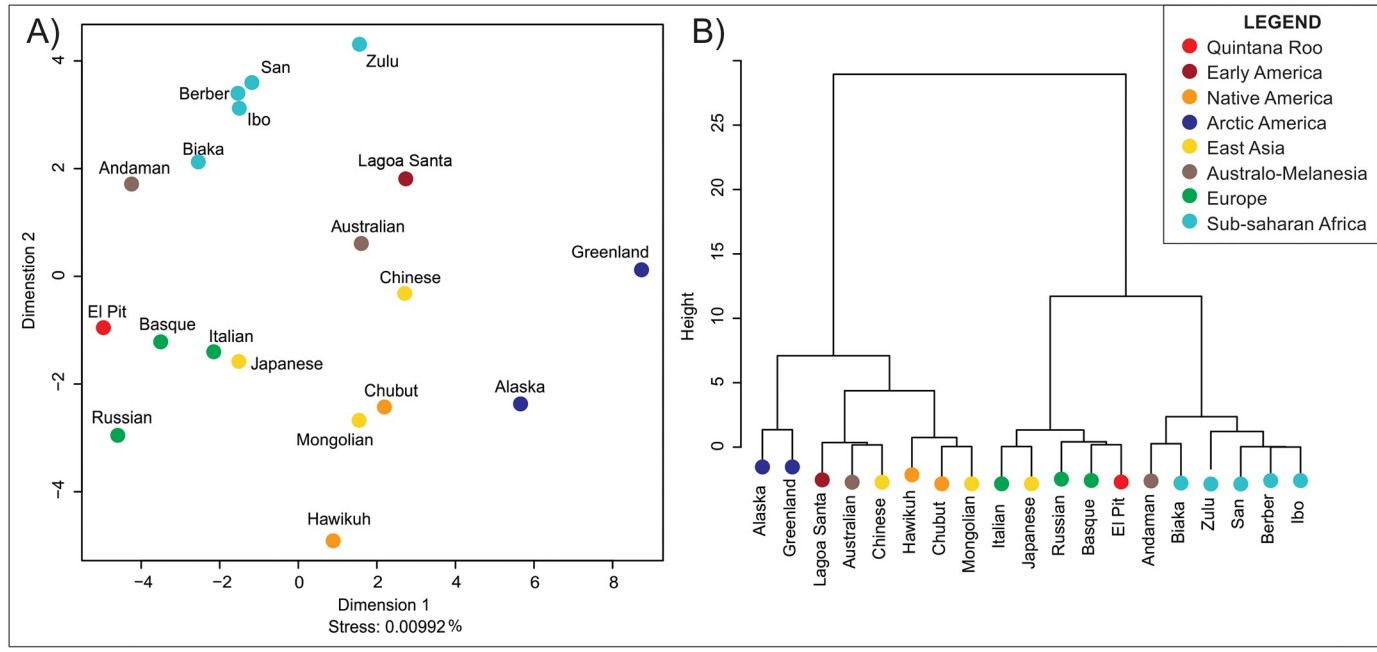

**Fig 4. Morphological affinities between El Pit I and reference series according to Mahalanobis distances of the first two Principal components.** A) Multidimensional Scaling. B) Ward's Cluster.

populations [65]). The Cluster analysis supports this association, showing that El Pit shares the larger cluster of European populations, and does not show any strong affinities with early or late American series. The classification analysis results (Table 4) shows that El Pit I has the highest Posterior Probabilities of being classified with the three European series and with Andaman. However, these probabilities never surpass 0.16, suggesting that El Pit I has relatively weak affinities with several series, rather than a strong affinity with one. The Typicality results (Table 4) support this interpretation, as only two of the reference series (Greenland and Alaska) can be considered statistically different (p<0.05) from this specimen, even though it is located close to the centroids (p>0.70) of the three European and Andaman series. El Pit I has a very different cranial vault morphology from the other Quintana Roo specimens (see SI1 for details), being the only individual that is brachiocephalic in the series. However, as was the case with Naharón, the morphological affinities of Pit I must be taken with caution, as the incomplete state of the specimen precludes any reliable conclusion of its relationship to the worldwide series.

## Las Palmas

The remaining two individuals from Quintana Roo (Las Palmas and Muknal) are considerably more complete, and as such should be considered as the most reliable specimens for the study of morphological affinities. Las Palmas (Fig 2) was estimated to be a mid-adult female at the time of her death, based on traits observable on the cranial and post-cranial skeleton. A complete description of the individual is presented in S1 Text. Las Palmas' chronology puts her at the boundary between the Pleistocene and Holocene, although the dates generated through AMS and U/Th show quite different time ranges (Table 1). The completeness of the individual allowed us to collect 37 landmarks in common with the reference dataset.

The morphological analyses for Las Palmas are based on the first five Principal Compo-
nents, which explain 54.4% of the variance in the dataset. The morphological affinities between
Las Palmas and the reference series is presented through the MDS analysis (Fig 5a) and the
Ward's Cluster (Fig 5b). With the increased number of variables contributing to the Mahala-
nobis distances, the MDS solution shows higher levels of stress (10.8%), but still within an
acceptable range [53]. The MDS analysis shows the same regional patterns observed before,
and Las Palmas appears strongly associated to the Paleoamerican series, suggesting strong
morphological affinities between the early American series included in this study. Interest-
ingly, both series appear in a central position in the plot, and the typical association between
Paleoamerican and Australian populations is not clear in this analysis. In fact, the Ward's Clus-
ter shows that the two series share a cluster with Asian and Native American series and show
no strong morphological affinities with the cluster from Africans and Australians. Finally, this
association with the Lagoa Santa series is evident in the classification analysis (Table 4), as Las
Palmas has its highest Posterior Probability of being part of the Lagoa Santa population
(p = 0.345), followed by smaller probabilities of being part of Chubut (p = 0.196) and Japanese
(p = 0.111). The Typicality shows that Las Palmas is very close to the Lagoa Santa centroid
(p = 0.882), but also have moderately high typicalities with Chubut (p = 0.717) and Japanese
(p = 0.546).

## Muknal

The last individual included in this study was recovered from Muknal and represents another
well-preserved skull. Muknal (Fig 2) is estimated to have been a male individual, who died
within the range of a mid-adult (30–45 years). A detailed description of this individual is pro-
vided in the S1 Text. Muknal shares the same chronological window as Las Palmas, with cali-
brated AMS dates ranging from 9,731 to 10,290 years BP (Table 1). Due to the good
preservation of the cranium, 27 landmarks in common with the reference series were collected
from this specimen.

The morphological analyses of Muknal are based on five Principal Components, which
explain 53.9% of the variance in the dataset. The morphological affinities of Muknal in com-
parison to the reference series can be observed in Fig 6a (MDS) and 6b (Ward's Cluster). The
MDS stress in this analysis is also within an acceptable range (8.4%). Different from Las Pal-
mas, Muknal does not show strong morphological affinities with Paleoamericans and appears
at the extreme of the morphological diversity in the MDS, with no clear pattern of morphologi-
cal affinities. On the Ward's Cluster analysis (Fig 6b), Muknal appears associated with the arc-
tic series, in a similar pattern to what is observed with the Naharón skull. This association,
however, is less strong than the one observed for Naharón, as the closest group to this individ-
ual in the Classification analysis is the South American series of Chubut (Table 4), with a Pos-
terior Probability of 0.317, followed by Alaska (p = 0.225) and Japanese (p = 0.106). However,
Muknal is the strongest outlier of all the Quintana Roo specimens, as indicated by the typical-
ity results (Table 4), where the highest typicality reported is quite low (0.406 for Chubut), and
ten of the reference series (all Africans, all Europeans and all Australo-Melanesians) are statis-
tically different (p<0.05) from this specimen. Therefore, the morphological associations of the
Muknal skull are not strongly defined, as the individual is a relatively strong outlier within the
morphospace shared by several of the American and Asian series in the reference dataset.

## Discussion and conclusions

Most of the studies that analyzed the morphological affinities of early Americans have shown
that these populations do not share strong morphological affinities with Native American

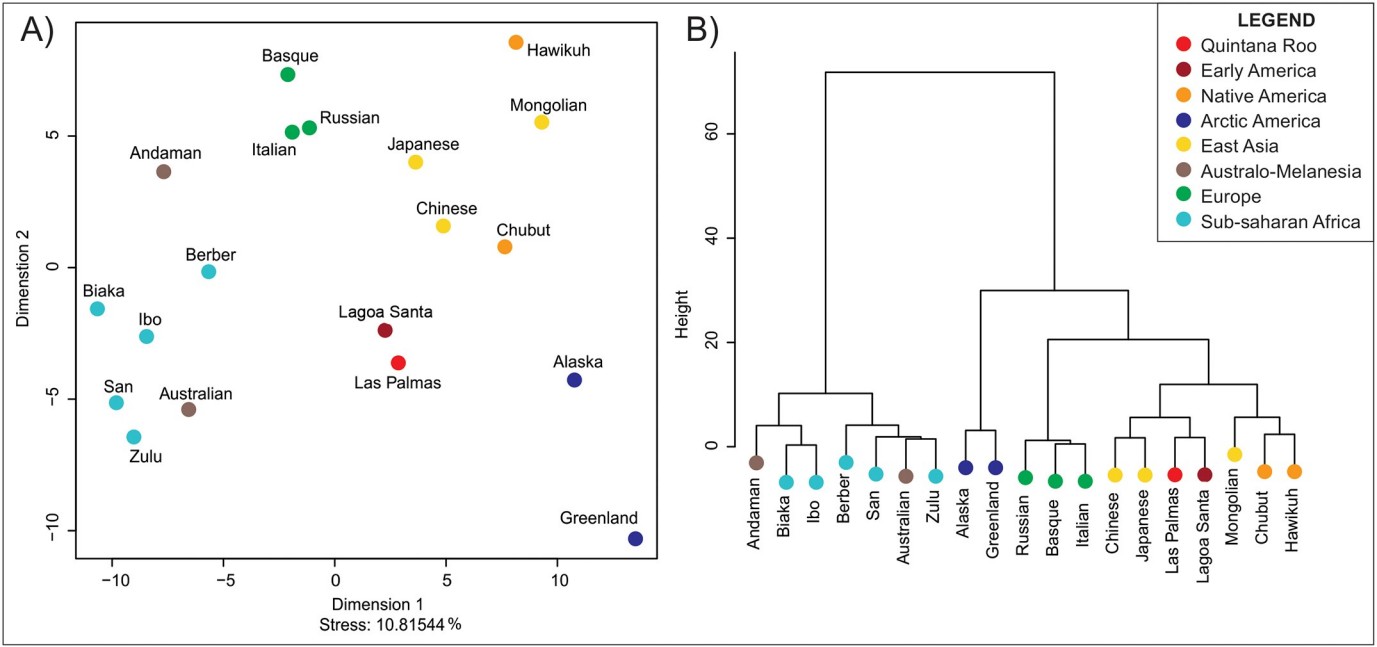

**Fig 5. Morphological affinities between Las Palmas and reference series according to Mahalanobis distances of the first five Principal components.** A) Multidimensional Scaling. B) Ward's Cluster.

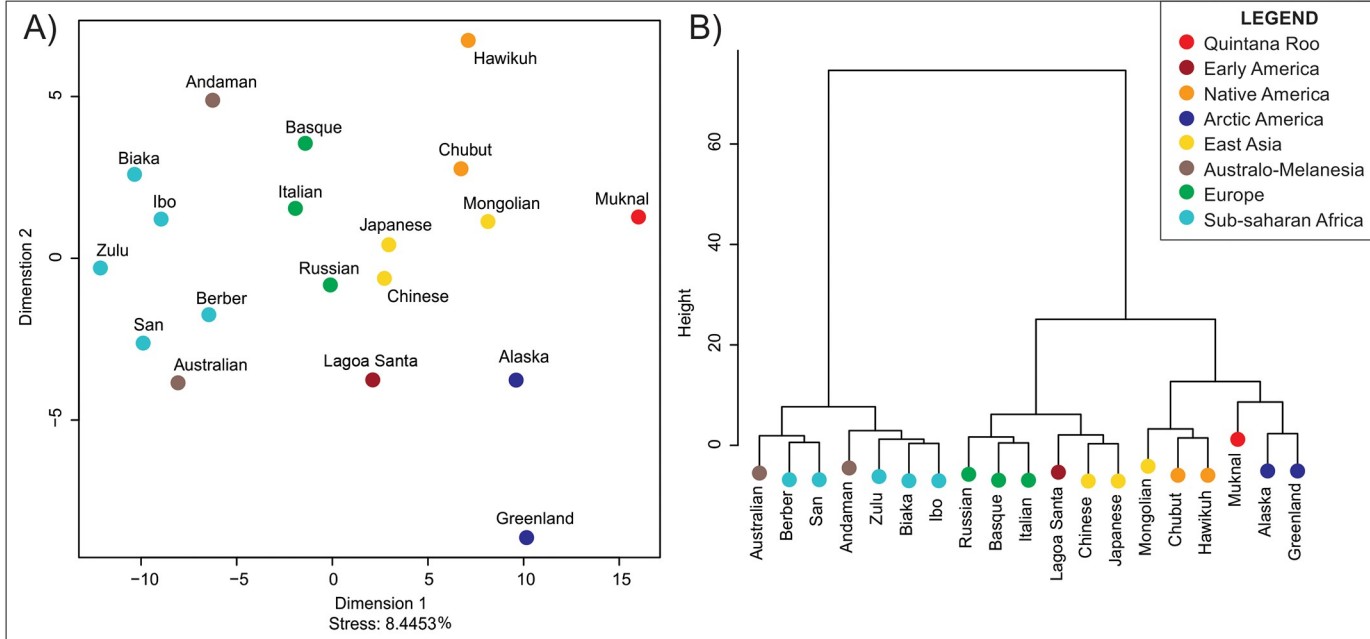

**Fig 6. Morphological affinities between Muknal and reference series according to Mahalanobis distances of the first five Principal components.** A) Multidimensional Scaling. B) Ward's Cluster.

populations [9, 10, 12, 13, 15, 16, 41, 63]. While there is immense debate about the reason for these differences (e.g., [9, 10, 15, 16, 56, 64]), there is substantial support that there is a significant shift in the cranial morphological pattern of Native American populations during the Holocene. Indeed, the morphological diversity seen in the continent over time is of the same magnitude as the difference observed between Australo-Melanesian and East Asian modern populations, which represent the most distinct regions on the planet in term of cranial morphological patterns [61, 62]. The early American morphology, commonly referred to as the Paleoamerican morphology [15, 63], seems to characterize most of the Native American populations during the last millennia of the Pleistocene and initial millennia of the Holocene [66]. After that, the morphological pattern that characterizes modern Native Americans becomes predominant, even though instances of the survival of the Paleomerican morphology have been reported across the continent [10, 38], including in the modern Mexican territory [40].

The early skulls from Quintana Roo fit well within this general pattern, as none (with a possible exception of Muknal) of them show a strong morphological affinity with more recent subarctic Native Americans series. As far as we can infer, based on the analysis of the individuals presented here, the early remains from the Quintana Roo follow the pattern of other early American series, in that they do not present a strong and evident association with later Native American series.

What distinguishes the Quintana Roo crania from other early American series is the degree of morphological diversity observed among them. In South America, where early American remains are more abundant, previous studies have shown a strong and consistent pattern of association with Australo-Melanesian and African series [10–12], as well as Late Pleistocene specimens from Europe and Asia [13]. This pattern has been explained as the result of early Americans retaining the ancestral morphology that characterized early modern human groups. For North America, the material available has shown less consistent patterns of morphological affinities [36, 65, 67, 68], although they have been described as having different morphological patterns from recent Native Americans [41, 68].

The Quintana Roo skulls do not seem to fit easily within the South American pattern, given that they show a remarkable degree of morphological diversity, each of them showing a different pattern of morphological affinities when compared to our reference series. Even though two of the specimens (Naharón and El Pit I) are very fragmented, and their morphological affinities should be considered less reliable, the two more complete skulls (Muknal and Las Palmas) show very different patterns of morphological affinities, suggesting that the observed morphological diversity is not just a result of the fragmented nature of the material. For most of the Quintana Roo skulls, we observe patterns of association that have been described before in the analysis of early American remains: Naharón and Muknal show a stronger affinity with North American arctic populations (Alaska and Greenland), which have been previously associated morphologically with early series from South America [16, 63]. Las Palmas also shows strong similarities with South American Paleoamericans, connecting this individual to the Paleoamerican morphological pattern [12, 38, 69]. As such, these crania demonstrate a strong affinity with populations that share a more generalized cranial morphology, as described in previous studies (e.g., [13]). The only exception to this is the individual from El Pit, which appears strongly associated with European series and shows a different overall cranial vault shape from the other Quintana Roo individuals (SI1). This pattern of association has not been observed before for South Paleoamericans (but see [65]), but some North American Early and Archaic skulls show stronger affinities with European series [41]. As such, the Quintana Roo specimens demonstrate an unexpected level of morphological diversity when compared with South Paleoamericans.

The high morphological diversity among the early Mexican material marks an interesting counterpoint to previous interpretations of early American diversity and as such has important implications for our understanding of the processes of early human movement across the continent. At the very least, it provokes researchers to reevaluate the validity of extrapolations made in the past. Since the beginning of the debate surrounding the settlement of the Americas, there has been a strong emphasis on grouping the human processes happening in the northern and southern continent as being similar. As such, most of the studies focusing on North America tend to assume that the occupation of South America can be extrapolated linearly from the northern continent (e.g., [17, 23, 25, 30]), while studies focusing on South America often assume that what is observed in that continent was also true for North America (e.g., [9, 15, 56, 63]), effectively ignoring the difference in archaeological evidence and eco-geographical realities between the two hemispherical regions for the sake of broad generalized models of human dispersion into the Americas.

The diversity seen in the Quintana Roo material suggests that, already at the Pleistocene/Holocene boundary, Native American individuals showed high morphological diversity, supporting studies of the few early specimens available from North America [41]. Moreover, this high diversity is not only restricted to the early populations, since recent groups in the Mexican region have also been described as presenting high morphological diversity [35, 41]. Together, these studies point to the Mexican territory being highly diverse across the entire time humans occupied it. In Baja California, the morphological affinities between the Pericues and Paleoamericans has been explained as the result of the former being isolated from other Native American populations during most of the Holocene [41]. However, in central Mexico, gene-flow barriers are not good explanations for the degree of morphological diversity reported by Herrera and colleagues [35], who suggest high diversity being present in the territory over longer periods of time.

It is hard to speculate at this point on the reasons why the Mexican territory would show a high degree of cranial morphological affinities over time, as this could be the result of a combination of different factors, including a) long-term gene-flow barriers between populations, established by either eco-geographical or cultural reasons; b) constant influx of new genetic diversity from the northern portion of the continent due to stronger gene-flow with those regions; and c) local processes of adaptation to different environmental conditions or life-style habits. Combinations of these different processes have been shown to be able to promote the appearance of morphological diversity among modern human populations over time [32, 33, 70–72]. Unfortunately, with only a few specimens available, it is impossible to test the contribution of any of these processes for the early Quintana Roo remains. However, while we cannot at this point contribute to the discussion on the possible origins of the high morphological diversity among early Mexican populations, establishing its presence allow us to put into question several aspects of the general views about the settlement of the Americas.

First, the high diversity in Quintana Roo, when compared to South American early remains, suggests that South America may have been occupied by groups carrying only a smaller portion of the total biological diversity in North America. This scenario is also supported by the study of early North Americans remains, which show different patterns of morphological affinities from what is observed in South America [41, 68]. This would explain the relative morphological homogeneity of South Paleoamericans, and also fits well with recent genetic data that shows Paleoamericans and other South American native groups share a common ancestor in North America [23]. Consequently, these results suggest that the abrupt change observed in the morphological pattern in South America does not need to be true in North America, and previous models of population replacement or multiple migration waves may only be applicable to South America.

Second, the Quintana Roo diversity supports previous studies that suggested similar levels of diversity in other parts of North America [41]. In other words, these individuals demonstrate that there is no reason to expect that all North American early individuals will share the Paleoamerican morphological pattern. In fact, this helps to contextualize some of the debated results found for some of the few early North American skulls, like Kennewick Man [65]. While this individual does not classify clearly with recent Native Americans, his morphology has been previously associated with Polynesian and European populations. It is important to clarify here that association with specific populations in the reference series does not imply a direct gene-flow or migration between them. In other words, strong morphological affinities between El Pit I (or other early North American specimens) with European populations does not imply that there was a migration from Europe to the Americas. It implies an unexpected level of morphological diversity, but it is not enough to establish ancestor-descendent relationships between reference series and the specimens analyzed.

Third, the identification of high morphological diversity among the Quintana Roo material, and even some of the unexpected morphological affinities between some of them and the reference series, demonstrates that we are still underestimating the degree of biological diversity observed in the continent. Bringing back the point made in the introduction of this article, until there is a reliable understanding of the biological diversity in the continent, broad spectrum models will always fall short in explaining the origins of Native American populations. As such, our results serve as a cautionary note to researchers building models based on evidence from only a few regions in the continent and encourage the continuous pursuit of new archaeological evidence of early populations in areas understudied in the continent.

Finally, our conclusions are based on the assumption that the individuals from Quintana Roo accurately represent the morphological diversity from their original population. While we were conscious of the fact that we are working with individuals, and adopted analytical strategies that try to control for the possible error caused by outlier individuals, it is hard to completely rule out that our overall results are being skewed by some of the individuals not being good representatives of their average population morphological pattern. For example, it has been suggested that Naharón possibly suffered from Klippel-Feil syndrome (see details in SI1), which may have produced variations in the shape of the skull that are unknown to us. Future analyses of this material should consider other sources of data to have a better understanding of the morphological relationships among the early populations of the continent. Detailed analyses of non-metric and other morphoscopic traits, like the morphology of the posterior portion of the skull, should be incorporated as the analysis of these specimens continues. For instance, Naharón and El Pit I present a common plane in the occipito-parietal suture and a pronounced supramastoid crest that extends beyond the parietomastoid suture over the posterior region of the temporal. Both individuals share a very developed mastoid process, which is associated to the formation of a distinct supramastoid sulcus. These traits not only show some degree of relationship between these specimens that cannot be easily assessed in a craniometric analysis like the one presented here, but they tie these specimens closely to other Paleomerican groups, as these traits are common in early specimens and absent in prehispanic Mayan populations (AT direct observation).

Despite their limitations, isolated specimens have always been important sources of information in the discussion of modern human expansion across the planet (e.g., [13, 67, 73]), contributing to the discussion of the pattern of mobility and migration between and within continents. As such, we believe that the material of Quintana Roo, even though fragmentary and represented by isolated specimens, is of special importance for the discussion about the processes of human occupation of the Americas, and that they allow us to propose new hypotheses and models, to be tested and refined with new findings in the future. At the very

least, these specimens represent now one of the best human remains collection known from the Pleistocene/Holocene boundary in North America, and they demonstrate patterns of morphological affinities that do not fit easily in our current models of human occupation of the continent.

## Supporting information

**S1 Table. Landmark coordinates for each Quintana Roo specimen.**
(XLSX)

**S1 Text. Detailed context of the Quintana Roo specimens.**
(DOCX)

## Author Contributions

**Conceptualization:** Mark Hubbe, Alejandro Terrazas Mata, Noreen Von Cramon-Taubadel.

**Data curation:** Alejandro Terrazas Mata, Brianne Herrera, Martha E. Benavente Sanvicente, Arturo González González, Carmen Rojas Sandoval, Jerónimo Avilés Olguín, Eugenio Acevez Núñez, Noreen Von Cramon-Taubadel.

**Formal analysis:** Mark Hubbe, Alejandro Terrazas Mata, Brianne Herrera, Noreen Von Cramon-Taubadel.

**Funding acquisition:** Alejandro Terrazas Mata.

**Investigation:** Alejandro Terrazas Mata, Martha E. Benavente Sanvicente, Arturo González González, Carmen Rojas Sandoval, Jerónimo Avilés Olguín, Eugenio Acevez Núñez.

**Methodology:** Brianne Herrera, Noreen Von Cramon-Taubadel.

**Project administration:** Alejandro Terrazas Mata.

**Visualization:** Mark Hubbe, Alejandro Terrazas Mata.

**Writing – original draft:** Mark Hubbe, Alejandro Terrazas Mata, Brianne Herrera.

**Writing – review & editing:** Mark Hubbe, Alejandro Terrazas Mata, Brianne Herrera, Noreen Von Cramon-Taubadel.

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
