## [Decision Letter · Decision Letter 0]

18 Sep 2019

PONE-D-19-23710

Morphological variation of the Early Human remains from Quintana Roo, Yucatán Peninsula, Mexico: contributions to the discussions about the settlement of the Americas

PLOS ONE

Dear Dr. Hubbe,

Thank you for submitting your manuscript to PLOS ONE. After careful consideration, we feel that it has merit but does not fully meet PLOS ONE’s publication criteria as it currently stands. Therefore, we invite you to submit a revised version of the manuscript that addresses the points raised during the review process.

Both reviewers thought that this paper had merit, and I would like to see it published in PLOS ONE.  Reviewer 1, in particular, provides a series of comments and recommendations that are reasonable.  I would recommend paying attention to recommendations for improvement of the Materials and Methods section and treatment of statistical results.  Reviewer 2 calls for referencing of a key publication and raises a point about further clarification and elaboration about expectations and outcomes.  

We would appreciate receiving your revised manuscript by Nov 02 2019 11:59PM. To enhance the reproducibility of your results, we recommend that if applicable you deposit your laboratory protocols in protocols.io, where a protocol can be assigned its own identifier (DOI) such that it can be cited independently in the future. For instructions see: http://journals.plos.org/plosone/s/submission-guidelines#loc-laboratory-protocols

We look forward to receiving your revised manuscript.

Kind regards,

Michael D. Petraglia, Ph.D.

Academic Editor

PLOS ONE

Journal Requirements:

3. In your manuscript, please provide additional information regarding the specimens used in your study. Ensure that you have reported specimen numbers and complete repository information, including museum name and geographic location.

For more information on PLOS ONE's requirements for paleontology and archaeology research, see https://journals.plos.org/plosone/s/submission-guidelines#loc-paleontology-and-archaeology-research.

4. We note that Figure 1 in your submission contain map/satellite images which may be copyrighted. All PLOS content is published under the Creative Commons Attribution License (CC BY 4.0), which means that the manuscript, images, and Supporting Information files will be freely available online, and any third party is permitted to access, download, copy, distribute, and use these materials in any way, even commercially, with proper attribution. For these reasons, we cannot publish previously copyrighted maps or satellite images created using proprietary data, such as Google software (Google Maps, Street View, and Earth). For more information, see our copyright guidelines: http://journals.plos.org/plosone/s/licenses-and-copyright.

You may seek permission from the original copyright holder of Figure 1 to publish the content specifically under the CC BY 4.0 license. 

If you are unable to obtain permission from the original copyright holder to publish these figures under the CC BY 4.0 license or if the copyright holder’s requirements are incompatible with the CC BY 4.0 license, please either i) remove the figure or ii) supply a replacement figure that complies with the CC BY 4.0 license. Please check copyright information on all replacement figures and update the figure caption with source information. If applicable, please specify in the figure caption text when a figure is similar but not identical to the original image and is therefore for illustrative purposes only.The following resources for replacing copyrighted map figures may be helpful:

Reviewers' comments:

Reviewer's Responses to Questions

**Comments to the Author**

1. Is the manuscript technically sound, and do the data support the conclusions?

Reviewer #1: Yes

Reviewer #2: Yes

2. Has the statistical analysis been performed appropriately and rigorously? 

Reviewer #1: Yes

Reviewer #2: Yes

3. Have the authors made all data underlying the findings in their manuscript fully available?

Reviewer #1: Yes

Reviewer #2: Yes

4. Is the manuscript presented in an intelligible fashion and written in standard English?

Reviewer #1: Yes

Reviewer #2: Yes

5. Review Comments to the Author

Reviewer #1: This research article presents an analysis of 3D geometric morphometric data collected from four late Pleistocene/early Holocene cranial specimens recovered from cave contexts in Quintana Roo, Mexico. Patterns of biological (morphological) distance and population affinity are generated and analyzed though use of an existing global reference sample. Specifically, the authors explore the relationship between individual specimens and the global samples with reference to Mahalanobis distances (Euclidian distances based on PC scores) via multidimensional scaling, hierarchical clustering, and classification (Posterior Probabilities and Typicalities). Results markedly differ from specimen to specimen, indicating a high degree of morphological diversity for early human occupants of Mexico. The authors conclude that current models of population origin, replacement, and migration are limited in that they significantly underestimate biological diversity in the Americas throughout the Pleistocene/Holocene transition.

This research article presents novel data on the early human occupation of Mexico. My overall impression of the paper is positive. The researchers clearly outline the importance of the Quintana Roo remains as some of the earliest dated on the continent and as geographically associated with a transcontinental “funnel” between North and South America. I believe the article is relevant to the journal’s readership and may be of interest to scholars of modern human migration histories, Paleoamerican population origins/dynamics, and global variation (esp. cranial shape/morphology). However, there are a few key issues the authors should address prior to publication. I recommend the research article be accepted once the authors make these revisions.

My recommendation is primarily based upon questions I have related to a) the presentation of global reference data, and b) MDS results as presented/interpreted. I would also ask that the authors clarify certain aspects of the study and results. Below I outline general and specific (primarily copy editing related) issues that the authors should address prior to resubmission.

General Comments:

1.Introduction, Page 2, Paragraph 2: I appreciate that the authors acknowledge the colonial and US-centric perspectives of early (and in many cases, current) academic pursuits of this topic. Introspection within our field, especially on these topics, is rarely presented in research articles, but hopefully that is changing.

2.Materials and Methods, Page 7, Paragraph 3 (and Table 3): The authors are interested in comparing the Quintana Roo sample to global series of varied temporal/biogeographic origin. The authors give general temporal/biogeographic “groupings” for these different samples, but do not discuss their composition in detail. This is not an issue for the more recent samples, but it may be necessary to outline the composition of the Paleoamerican series given the focus of this paper.

-Based on previous publications (and the paucity of Paleoamerican skeletal remains), it is likely the reference Paleoamerican series originates from one South American site. In their discussion, the authors mention limited morphological diversity of South Paleoamericans generally, but do not refer back to their reference sample specifically. Understanding the composition of the Paleoamerican series up-front would help to contextualize the results, which (for some skulls) are unexpected. However, if we understand the Paleoamerican sample to represent early human occupants of South America, and we can think of Mexico as a geographic “funnel” for population movement and migration patterns in the Americas, the results make more sense.

-Or perhaps this series includes Paleoamerican material from sites across North and South America—if so, the interpretation of the results would change. This is all to say, it would be helpful if you indicate the composition (site location/time period) of the Paleoamerican sample in the Materials and Methods section and/or Table 3.

3.Materials and Methods, Page 7, Paragraph 2: This paragraph is somewhat redundant in its justification for the use of PCA. Some of this paragraph can be condensed or cut. This discussion continues onto Page 8, Paragraphs 1 and 2. Again, some of this can be cut or condensed.

4.Results (Figures 3-6): The presented MDS plots include stress values as a measure of goodness of fit. I assume these are Kruskal stress values, but it is unclear (especially from the discrepancies from Figures 3-4 to Figures 5-6, whether these are presented as raw values or percentages. If they are presented as raw values, my concern is for the loss of information in the Figure 5 and 6 MDS plots. Optimally, stress values should fall below 0.15, although there is no hard and fast rule. Even if the stress value is high, my concern is that the values are so different across the four analyses, making direct comparison of MDS-based results difficult (the relationships presented in the Figure 5 and 6 MDS are almost certainly distorted relative to those in the Figure 3 and 4 MDS).

-I would first recommend that the authors explicitly state their goodness of fit measure and how it is presented (raw value or percentage).

-If this is a raw Kruskal stress value, I would recommend the authors increase dimensionality of the Figure 5 and 6 plots to improve fit. A 3D MDS plot might provide a more accurate reflection of the relationships in the two final analyses.

-At a minimum, I would acknowledge the poor fit of the Figure 5 and 6 MDS output, as well as the related interpretive limitations.

5.Discussion, Page 17, Paragraph 1: The authors mention that Naharón may have suffered from Klippel-Feil syndrome. This seems to come out of nowhere at the end of the paper and is not accompanied by a reference to previous studies and/or the supplemental files. After reviewing the supplemental material, I realized it was mentioned, but I would suggest adding a reference to the supplemental files when first mentioning it in the text of the paper.

Specific Comments:

1.Abstract, Page 1: Please change the phrase “…already shared a higher degree…” to “already shared a high degree…”.

2.Materials and Methods, Page 6, Paragraph 2: Please change the sentence “While this limits our ability to explore the affinities among the Quintana Roo specimens, such a comparison…” to “This limits our ability to explore the affinities among the Quintana Roo specimens; such a comparison…”.

3.Materials and Methods, Page 8, Paragraph 2: Please change the phrase “…for each of the analyses done we chose…” to “…for each of the analyses we chose…”.

4.Materials and Methods, Page 8, Paragraph 2: When explaining why Mahalanobis distances calculated from PC scores are technically Euclidean distances, perhaps it would be useful to indicate in the paragraphs preceding that PCA results in orthogonal, uncorrelated variables (thus the variance of 1, covariances of 0).

5.Materials and Methods, Page 8, Paragraph 2: There is no mention of goodness of fit tests for multidimensional scaling. From the figures it appears that stress values were referenced. I am assuming these were Kruskal stress values, but no “cut off” for fit is mentioned. This brings me to my more general comment listed above.

6.Materials and Methods, Page 9, Paragraph 1: The sentence that begins “In all analyses, males and females…” is grammatically incorrect. Please edit this sentence.

7.Results, Page 10, Paragraph 1: Please change the phrase “The Typicality results show Naharón to be very close from the centroid…” to “The Typicality results show Naharón to be very close to the centroid…”.

Reviewer #2: There is no citation to Jantz and Owsley (2001; AJPA) who perform a similar analysis of 11 Paleoindian/early archaic skulls. They even do a somewhat similar analysis, calculating distances of each skull from worldwide samples and providing typicality probabilities along with the distance. Their findings support your argument. You should acknowledge this. Also, even though dealing with recent skeletons, WW Howells often found Native Americans clustered with Europeans, not Asians or Pacific populations. In general, my students and I have done similar analyses and often find strange assignments based on craniometric dimensions. I'm unclear as to how your findings diverge from your expectations. Some elaboration on that point would help the reader.

6. PLOS authors have the option to publish the peer review history of their article (what does this mean?). If published, this will include your full peer review and any attached files.

Reviewer #1: No

Reviewer #2: No

---

## [Author Response · Author response to Decision Letter 0]

11 Dec 2019

RESPONSE TO REVIEWERS:

Reviewer #1: This research article presents an analysis of 3D geometric morphometric data collected from four late Pleistocene/early Holocene cranial specimens recovered from cave contexts in Quintana Roo, Mexico. Patterns of biological (morphological) distance and population affinity are generated and analyzed though use of an existing global reference sample. Specifically, the authors explore the relationship between individual specimens and the global samples with reference to Mahalanobis distances (Euclidian distances based on PC scores) via multidimensional scaling, hierarchical clustering, and classification (Posterior Probabilities and Typicalities). Results markedly differ from specimen to specimen, indicating a high degree of morphological diversity for early human occupants of Mexico. The authors conclude that current models of population origin, replacement, and migration are limited in that they significantly underestimate biological diversity in the Americas throughout the Pleistocene/Holocene transition.

This research article presents novel data on the early human occupation of Mexico. My overall impression of the paper is positive. The researchers clearly outline the importance of the Quintana Roo remains as some of the earliest dated on the continent and as geographically associated with a transcontinental “funnel” between North and South America. I believe the article is relevant to the journal’s readership and may be of interest to scholars of modern human migration histories, Paleoamerican population origins/dynamics, and global variation (esp. cranial shape/morphology). However, there are a few key issues the authors should address prior to publication. I recommend the research article be accepted once the authors make these revisions.

My recommendation is primarily based upon questions I have related to a) the presentation of global reference data, and b) MDS results as presented/interpreted. I would also ask that the authors clarify certain aspects of the study and results. Below I outline general and specific (primarily copy editing related) issues that the authors should address prior to resubmission.

RESPONSE: WE TRULY APPRECIATE THE SUGGESTIONS MADE BY THE REVIEWER AND HER/HIS SUPPORT FOR OUR WORK. WE ADDRESSED ALL THE SUGGESTIONS MADE, AND WE BELIEVE THE MANUSCRIPT IS CLEARER THANKS TO THE SUGGESTIONS MADE.

General Comments:

1.Introduction, Page 2, Paragraph 2: I appreciate that the authors acknowledge the colonial and US-centric perspectives of early (and in many cases, current) academic pursuits of this topic. Introspection within our field, especially on these topics, is rarely presented in research articles, but hopefully that is changing.

RESPONSE: WE APPRECIATE THE COMMENT BY THE REVIEWER, AND AGREE THAT THIS IS AN ASPECT OF THE STUDY OF THE PAST THAT SHOULD BECOME MORE EXPLICIT IN OUR STUDIES.

2.Materials and Methods, Page 7, Paragraph 3 (and Table 3): The authors are interested in comparing the Quintana Roo sample to global series of varied temporal/biogeographic origin. The authors give general temporal/biogeographic “groupings” for these different samples, but do not discuss their composition in detail. This is not an issue for the more recent samples, but it may be necessary to outline the composition of the Paleoamerican series given the focus of this paper.

-Based on previous publications (and the paucity of Paleoamerican skeletal remains), it is likely the reference Paleoamerican series originates from one South American site. In their discussion, the authors mention limited morphological diversity of South Paleoamericans generally, but do not refer back to their reference sample specifically. Understanding the composition of the Paleoamerican series up-front would help to contextualize the results, which (for some skulls) are unexpected. However, if we understand the Paleoamerican sample to represent early human occupants of South America, and we can think of Mexico as a geographic “funnel” for population movement and migration patterns in the Americas, the results make more sense.

-Or perhaps this series includes Paleoamerican material from sites across North and South America—if so, the interpretation of the results would change. This is all to say, it would be helpful if you indicate the composition (site location/time period) of the Paleoamerican sample in the Materials and Methods section and/or Table 3.

RESPONSE: THE ENTIRE SERIES OF PALEOAMERICANS CONSISTS OF INDIVIDUALS FROM THE LAGOA SANTA REGION IN BRAZIL, MEASURED BY NVCT. THIS REPRESENTS THE LARGEST AND PROBABLY ONLY COLLECTION OF EARLY AMERICANS IN THE CONTINENT THAT PERMIT THE INFERENCE OF POPULATIONAL PARAMETERS. WE CLARIFIED THIS IN THE MANUSCRIPT, AND CHANGED FIGURES AND TABLES, WHICH NOW PRESENT THIS SERIES AS LAGOA SANTA, INSTEAD OF PALEOAMERICANS. 

3.Materials and Methods, Page 7, Paragraph 2: This paragraph is somewhat redundant in its justification for the use of PCA. Some of this paragraph can be condensed or cut. This discussion continues onto Page 8, Paragraphs 1 and 2. Again, some of this can be cut or condensed.

RESPONSE: THE PARAGRAPH HAS BEEN CONDENSED AND REDUNDACIES REMOVED.

4.Results (Figures 3-6): The presented MDS plots include stress values as a measure of goodness of fit. I assume these are Kruskal stress values, but it is unclear (especially from the discrepancies from Figures 3-4 to Figures 5-6, whether these are presented as raw values or percentages. If they are presented as raw values, my concern is for the loss of information in the Figure 5 and 6 MDS plots. Optimally, stress values should fall below 0.15, although there is no hard and fast rule. Even if the stress value is high, my concern is that the values are so different across the four analyses, making direct comparison of MDS-based results difficult (the relationships presented in the Figure 5 and 6 MDS are almost certainly distorted relative to those in the Figure 3 and 4 MDS).

-I would first recommend that the authors explicitly state their goodness of fit measure and how it is presented (raw value or percentage).

-If this is a raw Kruskal stress value, I would recommend the authors increase dimensionality of the Figure 5 and 6 plots to improve fit. A 3D MDS plot might provide a more accurate reflection of the relationships in the two final analyses.

-At a minimum, I would acknowledge the poor fit of the Figure 5 and 6 MDS output, as well as the related interpretive limitations.

RESPONSE: WE CLARIFIED THAT WE ARE USING KRUSKAL’S STRESS AS A MEASUREMENT OF GOODNESS OF FIT. THIS IS EXPLAINED IN M&M, ON PAGE 9 OF THE REVISED MANUSCRIPT. HOWEVER, WE DID NOT PRESENT 3D MDS SOLUTIONS FOR THE MOST COMPLETE SKULLS, FOR TWO REASONS:

1. THE STRESS LEVEL FOR THE 2D MDS STILL FALLS WITHIN THE RANGE OF WHAT IS CONSIDERED A FAIR REPRESENTATION OF THE DISTANCE MATRIX (~10%). THIS HAS BEEN EXPLAINED ON PAGE 9 OF THE NEW MANUSCRIPT AS WELL.

2. THE 3D MDS SOLUTION, WHILE REDUCING STRESS LEVEL TO ~5% ON BOTH ANALYSIS, DID NOT CHANGE SIGNIFICANTLY THE AFFINITIES OBSERVED IN THE 2D SOLUTION. BECAUSE THE REPRESENTATION OF 3D PLOTS ON 2D SPACE IS ALWAYS AWKWARD, WE CHOSE TO MAINTAIN THE 2D SOLUTION, TO BE CONSISTENT WITH THE OTHER ANALYSES.

WE ALSO WANT TO NOTE THAT THE DIFFERENCE IN STRESS LEVELS IS A RESULT OF THE NUMBER OF PCS CONTRIBUTING TO THE DISTANCE MATRIX. AS THE FIRST TWO SPECIMENS ARE REPRESENTED BY ONLY TWO DIMENSIONS (PCS) IN THE DISTANCE MATRIX, THE STRESS OF THE 2D MDS IS NEAR ZERO. THIS HAS BEEN CLARIFIED IN THE RESULTS AS WELL (SEE ADDITIONS TO RESULTS, PAGES 10-13 OF THE REVISED MANUSCRIPT).

5.Discussion, Page 17, Paragraph 1: The authors mention that Naharón may have suffered from Klippel-Feil syndrome. This seems to come out of nowhere at the end of the paper and is not accompanied by a reference to previous studies and/or the supplemental files. After reviewing the supplemental material, I realized it was mentioned, but I would suggest adding a reference to the supplemental files when first mentioning it in the text of the paper.

RESPONSE: WE FOLLOWED THE REVIEWER’S SUGGESTION AND REFERRED THE READER TO THE SUPPLEMENTARY INFORMATION.

Specific Comments:

RESPONSE: ALL SUGGESTIONS BELOW HAVE BEEN INCORPORATED INTO THE MANUSCRIPT. ONCE MORE, WE THANK THE REVIEWER FOR THE CAREFUL READ OF THE MANUSCRIPT. 

1.Abstract, Page 1: Please change the phrase “…already shared a higher degree…” to “already shared a high degree…”.

done

2.Materials and Methods, Page 6, Paragraph 2: Please change the sentence “While this limits our ability to explore the affinities among the Quintana Roo specimens, such a comparison…” to “This limits our ability to explore the affinities among the Quintana Roo specimens; such a comparison…”.

3.Materials and Methods, Page 8, Paragraph 2: Please change the phrase “…for each of the analyses done we chose…” to “…for each of the analyses we chose…”.

4.Materials and Methods, Page 8, Paragraph 2: When explaining why Mahalanobis distances calculated from PC scores are technically Euclidean distances, perhaps it would be useful to indicate in the paragraphs preceding that PCA results in orthogonal, uncorrelated variables (thus the variance of 1, covariances of 0).

5.Materials and Methods, Page 8, Paragraph 2: There is no mention of goodness of fit tests for multidimensional scaling. From the figures it appears that stress values were referenced. I am assuming these were Kruskal stress values, but no “cut off” for fit is mentioned. This brings me to my more general comment listed above.

6.Materials and Methods, Page 9, Paragraph 1: The sentence that begins “In all analyses, males and females…” is grammatically incorrect. Please edit this sentence.

7.Results, Page 10, Paragraph 1: Please change the phrase “The Typicality results show Naharón to be very close from the centroid…” to “The Typicality results show Naharón to be very close to the centroid…”.

Reviewer #2: There is no citation to Jantz and Owsley (2001; AJPA) who perform a similar analysis of 11 Paleoindian/early archaic skulls. They even do a somewhat similar analysis, calculating distances of each skull from worldwide samples and providing typicality probabilities along with the distance. Their findings support your argument. You should acknowledge this. Also, even though dealing with recent skeletons, WW Howells often found Native Americans clustered with Europeans, not Asians or Pacific populations. In general, my students and I have done similar analyses and often find strange assignments based on craniometric dimensions. I'm unclear as to how your findings diverge from your expectations. Some elaboration on that point would help the reader.

RESPONSE: WE ARE THANKFUL FOR THE REFERENCE PROVIDED BY THE REVIEWER. INDEED JANTZ AND OWSLEY IS AN IMPORTANT REFERENCE THAT WE MISSED. THE REFERENCE WAS INTRODUCED AND WE UPDATED THE DISCUSSION TO INCORPORATE A BROADER PICTURE OF THE NORTH AMERICAN EARLY CRANIAL DIVERSITY. PLEASE REFER TO THE CHANGES MADE TO THE DISCUSSION (PAGES 14-18 OF THE REVISED MANUSCRIPT).

AS THIS STUDY IS FOCUSED ON AN EXPLORATORY EVALUATION OF THE QUINTANA ROO MATERIAL, WE AVOIDED BUILDING UP EXPLICIT EXPECTATIONS FOR THE MATERIAL (IN LIGHT OF THE POINTS MADE ON THE INTRODUCTION AS WELL), REASON WHY WE DID NOT DEVELOP FURTHER ON HOW RESULTS DIVERGE FROM EXPECTATION. WE HOPE, HOWEVER, THAT THE NEW ADDITIONS TO THE DISCUSSION BRING THE RESULTS TO A BROADER CONTEXT OF MEANING.

---

## [Decision Letter · Decision Letter 1]

19 Dec 2019

Morphological variation of the early Human remains from Quintana Roo, Yucatán Peninsula, Mexico: contributions to the discussions about the settlement of the Americas

PONE-D-19-23710R1

Dear Dr. Hubbe,

We are pleased to inform you that your manuscript has been judged scientifically suitable for publication and will be formally accepted for publication once it complies with all outstanding technical requirements.

With kind regards,

Michael D. Petraglia, Ph.D.

Academic Editor

PLOS ONE

Additional Editor Comments (optional):

Reviewers' comments:

Reviewer's Responses to Questions

**Comments to the Author**

1. If the authors have adequately addressed your comments raised in a previous round of review and you feel that this manuscript is now acceptable for publication, you may indicate that here to bypass the “Comments to the Author” section, enter your conflict of interest statement in the “Confidential to Editor” section, and submit your "Accept" recommendation.

Reviewer #1: All comments have been addressed

2. Is the manuscript technically sound, and do the data support the conclusions?

Reviewer #1: Yes

3. Has the statistical analysis been performed appropriately and rigorously? 

Reviewer #1: Yes

4. Have the authors made all data underlying the findings in their manuscript fully available?

Reviewer #1: Yes

5. Is the manuscript presented in an intelligible fashion and written in standard English?

Reviewer #1: Yes

6. Review Comments to the Author

Reviewer #1: I thank the authors for their thoughtful responses to my comments. I recommend the paper be accepted for publication.

I thank the Editor for the opportunity to review this paper, and look forward to serving the journal in the future.

7. PLOS authors have the option to publish the peer review history of their article (what does this mean?). If published, this will include your full peer review and any attached files.

Reviewer #1: No

---

## [Editor Report · Acceptance letter]

7 Jan 2020

PONE-D-19-23710R1 

Morphological variation of the early Human remains from Quintana Roo, Yucatán Peninsula, Mexico: contributions to the discussions about the settlement of the Americas 

Dear Dr. Hubbe:

I am pleased to inform you that your manuscript has been deemed suitable for publication in PLOS ONE. Congratulations! Your manuscript is now with our production department. 

With kind regards,

on behalf of

Professor Michael D. Petraglia 

Academic Editor

PLOS ONE